# Automatic Fetal Middle Sagittal Plane Detection in Ultrasound Using Generative Adversarial Network

**DOI:** 10.3390/diagnostics11010021

**Published:** 2020-12-24

**Authors:** Pei-Yin Tsai, Ching-Hui Hung, Chi-Yeh Chen, Yung-Nien Sun

**Affiliations:** 1Department of Obstetrics and Gynecology, National Cheng Kung University Hospital, College of Medicine, National Cheng Kung University, Tainan 70104, Taiwan; tsaipy@mail.ncku.edu.tw; 2Institute of Computer Science and Information Engineering & Institute of Medical Informatics, National Cheng Kung University, Tainan 70104, Taiwan; doggyching@gmail.com

**Keywords:** automatic detection, middle sagittal plane (MSP), generative adversarial network (GAN), three-dimensional (3D), ultrasound (US)

## Abstract

Background and Objective: In the first trimester of pregnancy, fetal growth, and abnormalities can be assessed using the exact middle sagittal plane (MSP) of the fetus. However, the ultrasound (US) image quality and operator experience affect the accuracy. We present an automatic system that enables precise fetal MSP detection from three-dimensional (3D) US and provides an evaluation of its performance using a generative adversarial network (GAN) framework. Method: The neural network is designed as a filter and generates masks to obtain the MSP, learning the features and MSP location in 3D space. Using the proposed image analysis system, a seed point was obtained from 218 first-trimester fetal 3D US volumes using deep learning and the MSP was automatically extracted. Results: The experimental results reveal the feasibility and excellent performance of the proposed approach between the automatically and manually detected MSPs. There was no significant difference between the semi-automatic and automatic systems. Further, the inference time in the automatic system was up to two times faster than the semi-automatic approach. Conclusion: The proposed system offers precise fetal MSP measurements. Therefore, this automatic fetal MSP detection and measurement approach is anticipated to be useful clinically. The proposed system can also be applied to other relevant clinical fields in the future.

## 1. Introduction

Ultrasound (US), as a convenient, powerful, and effective tool, is widely used for prenatal growth assessment and plays an important role in prenatal diagnosis. With the rapid development of US technology, the inspection results are becoming more detailed and clearer. Most major fetal abnormalities can be identified by US before delivery, even in the first trimester of pregnancy [1]. In addition to structural assessments, certain indicators can be used to screen for chromosomal abnormalities [2]. Few unexpected findings and some major structural abnormality with thick nuchal translucency could be identified in first trimester scans of patients with negative cell-free DNA [3,4]. Furthermore, early scanning for fetal congenital anomalies is an essential component of modern pregnancy care in the cell-free DNA era [5]. However, accurate US inspections require highly skilled professionals with appropriate training, since US image quality may be affected by speckle noise, fuzzy boundaries, and weak edges. Unsatisfactory results can lead to erroneous conclusions, medical waste, and unnecessary anxiety. Three-dimensional (3D) US is valuable in prenatal diagnosis of fetal structures because it provides a multi-planar view [6,7]. Although 3D US has improved the visibility of the fetal structure, discrimination between normal and abnormal structures remains difficult and depends on expert judgement. The acquisition of a true middle sagittal plane (MSP) of the fetus is the fundamental prerequisite for reliable measurement and the basis for the nuchal translucency exam that provides a risk assessment for chromosomal aberrations in the first trimester [8]. The ideal plane is the main requirement for obtaining effective and repeatable measurements and maintaining inspection quality [9]. Fetal structural measurements require that an expert obtain a standard plane, which is time consuming and subjective [10]. Automated systems can increase efficiency, reliability, and accuracy in clinical medicine applications [11]. Automated systems are quite popular in the medical field and have been successfully used for many years in US applications. There are semi-automatic/automatic systems for fetal assessment in US imaging [12,13,14,15,16,17,18]. Considering that more challenging modes and image recognition processes have been implemented, the use of automation in medical and US applications is logical and feasible. Therefore, using image analysis technology, we developed an automated system using deep learning with a generative adversarial network (GAN) for MSP detection.

## 2. Materials and Methods

This study was approved by the Institutional Review Board of National Cheng Kung University Hospital (NCKUH, No.: B-ER-102-402 was approved on 6 July 2016). Women with normal pregnancy at gestational ages of 11–13 weeks were recruited from the antenatal outpatient department of National Cheng Kung University Hospital. Only women without maternal diseases known to affect fetal growth, i.e., pre-existing hypertension or diabetes mellitus, and pregnancies that were not at risk for fetal abnormalities were included in the study, after the study was approved and informed consent was obtained. The pregnancy duration was determined from the last reliable menstrual period or, in case of uncertainty, adjusted by US in the early first trimester of gestation. Women with singleton pregnancies resulting in the term delivery of an infant without congenital anomalies were recruited.

The whole fetus volumes were acquired using a trans-abdominal 3D transducer with a frequency range of 4–8 MHz (Voluson 730Expert and E8, GE Healthcare, Kretz Ultrasound, Zipf, Austria). The acquisition angle, which is 85° in most volumes, was set to ensure the inclusion of the entire gestational sac and fetus. The image volumes were acquired by the appropriate training of sonographers and adherence to a standard technique in accordance with the guidelines that were established by The Fetal Medicine Foundation (FMF). The guidelines of 3D US include the whole fetus (fetal crown-rump length should be obtained), the fetus is in the neutral position, and the amnion is seen separately from the nuchal membrane. The proposed framework was developed using Python on an Intel i7 CPU (3.2 GHz, 6 cores) and training was performed on a single NVIDIA 1080Ti GPU with the Tensor flow library from scratch. The fetal MSP was detected automatically using the software and manually by the two obstetrics doctors with 20 years (Pei-Yin Tsai MD.) and 6 years (Pei-Hsiu Yu MD.) of experience.

The proposed system is a two-stage deep learning method. In the first stage, deep learning is used to find a seed point for the fetal head. In the second stage, a GAN is utilized for MSP detection in 3D fetal US images. According to the four anatomical features (nuchal translucency, nasal tip, nasal bone, and diencephalon) of the standard fetal MSP in a 3D US image, the objective of MSP detection is to search one plane from a volume that exactly splits the fetus into right and left halves with crossing of the requiring features. The proposed method learns not only the specific feature, but also the position information simultaneously.

In the first stage, a deep learning method for finding a seed point of the fetal head is utilized. In total, four deep learning methods are employed to obtain an exact seed point. The segmentation network firstly finds the seed point in the sagittal view and then obtains its location (*x*, *y*). After finding (*x*, *y*), two object detection networks are used to identify location *z* in the axial and coronal views. Then, the first segmentation network is utilized to refine (*x*, *y*). Finally, according to the location *y*, another segmentation network refines the location *z* in the axial view.

In the second stage, a deep learning method involving a GAN, which contains a generator and discriminator, is used for automatic fetal MSP detection in 3D US images. In the work of WGAN, Arjovsky et al. [19] rename the discriminator to critic for emphasizing its property. In this paper, we also called the discriminator as a critic. The generator input was a cropped volume and the output was a 3D binary mask, where the input and output have the same sizes. The MSP position information was embedded in the 3D mask, where the value was one if the voxel is included in the MSP and zero otherwise. The input of the critic was a combination of a 3D mask and image data with a combination operation. The combination operation multiplied the predicted 3D mask and input image element by element to obtain the intensity plane from the original image. Then, it concatenated the result of multiplication with the input image. Hence, the output of the combination operation was two-channel data (Figure 1).

## 3. GAN for MSP Detection in 3D Fetal US Images

This section firstly proposes a deep learning method for finding a seed point of the fetal head. Then, it proposes a GAN for MSP detection in 3D fetal US images. According to four anatomical features of the standard MSP in a 3D fetal US image, the goal of MSP detection is to search one plane from a volume that exactly splits the fetus into the right and left halves with crossing of the requiring features. An instinctive idea is to classify all possible slices as true or false according to the similarity to the ground truth plane. However, classifying a large number of planes is very time consuming. Moreover, judging the comparisons only based on 2D images causes the loss of location information of the planes with respect to the fetus in 3D space. Therefore, MSP detection was treated as filtration in this work to overcome the issues. That is, we employed a neural network to find a seed point of the fetal head and generate a 3D binary mask. The proposed method learns not only the specific features, but also the position information simultaneously.

### 3.1. Deep Learning Method for Finding a Seed Point of a Fetal Head

This section proposes a deep learning method for finding a seed point of a fetal head, in which four deep learning networks are employed to obtain an exact seed point. Two segmentation networks in the Unet + ASPP [20] architecture (see Figure 2) are utilized for the sagittal and axial views, and two additional networks are used for object detection and obtain the seed point from the axial and coronal views. The atrous spatial pyramid pooling (ASPP) probes an incoming convolutional feature layer with filters at multiple sampling rates and effective fields-of-views, thus capturing objects as well as image context at multiple scales [21]. The two object detection networks are deep learning networks that are used to modify the predicted seed point. The detection procedure is as follows. Firstly, the segmentation network finds the seed point in the sagittal view and then obtains its location (*x*, *y*). After finding (*x*, *y*), the two object detection networks are employed to find the location *z* in the axial and coronal views. Then, the first segmentation network is used to refine (*x*, *y*). Finally, according to the location *y*, another segmentation network is utilized to refine the location *z* in the axial view.

### 3.2. Overview of the GAN-Based Fetal MSP Detection Approach

This section proposes a deep learning method for automatic MSP detection in 3D fetal US images. The proposed method is based on the GAN shown in Figure 3, which contains a generator and critic.

The input of the generator is a cropped volume, and its output is a 3D binary mask, where the input and output have the same size. The MSP position information is embedded in the 3D mask, where the value is one if the voxel is included in the MSP, zero otherwise. The input of the critic are data from a combination operation that multiplies the predicted 3D mask times the input image element by to obtain the intensity plane from the original image. Then, it concatenates the multiplication results with the input image. Hence, the output of the combination operation is two-channel data.

In the testing phase, only the generator is used to predict a 3D binary mask. In post-processing, the 3D binary mask is processed with the original input image and the final 2D MSP image is obtained (see Figure 4).

### 3.3. Network Architecture

#### 3.3.1. Generator

The generator is a symmetric 3D autoencoder, as shown in Figure 5. The encoder is composed of four convolutional layers. After two fully connected layers with leaky ReLU layers, the decoder includes four deconvolution layers. A leaky ReLU layer is employed after every deconvolution layer, except for the last layer, where a sigmoid layer is used instead.

#### 3.3.2. Critic

The architecture of the critic is similar to the encoder of the generator and contains four convolutional layers. Each layer is followed by a leaky ReLU layer except for the last layer, where a sigmoid layer is utilized. In addition, a max-pooling layer is used in every layer. The number of output channels is the same as that in the encoder of the generator, as shown in Figure 6. It is worth noting that the output of the critic is a latent vector, instead of a value, representing the distribution of a real or fake mask.

### 3.4. Loss

In the original GAN, the loss function is the Jensen–Shannon divergence, which makes it difficult to achieve convergence in training. To overcome this issue, Wasserstein distance-based loss function with weight clipping (WGAN) was proposed. In the extended version of WGAN, namely, WGAN-GP, weight clipping is replaced with a gradient penalty with respect to the input of the critic. Based on WGAN-GP, the filter weights of two networks were trained on a pair of loss functions in this work.

Let *L^G^* and *L^C^* be the loss functions for updating the generator and critic. The loss function *L^G^* has a cross-entropy term *L^ce^*, which is not present in the original loss function of the generator in WGAN-GP. The cross-entropy term can make the prediction and ground truth as similar as possible. Let *y* be the ground truth mask, *x* be the predicted output mask, and $\hat{x}=\alpha y+\left(1-\alpha \right)x$ be a linear combination of *x* and *y* with a random weight α ∈ (0, 1). Let *x*′, *y*′, and ${\hat{x}}^{\prime }$ be the inputs of the critic after the combination of the generator input and *x*, *y*, and $\hat{x}$, respectively. Hence, the two loss functions *L^G^* and *L^C^* are
(1)$$L^{G}=-\left(1-w\right)\left(E\left[C\left(x^{\prime }\right)\right]\right)+wL^{ce}$$
(2)$$L^{C}=E\left[C\left(x^{\prime }\right)\right]-E\left[C\left(y^{\prime }\right)\right]+\lambda E\left[{\left(\|{\nabla_{{\hat{x}}^{\prime }}C\left({\hat{x}}^{\prime }\right)\|}_{2}-1\right)}^{2}\right]$$
where *L^ce^ = **E*** [−*y*log (*x*)−(1−*y*) log (1−*x*)], *C* is the critic, *E* is the expectation, λ is a weight for the gradient penalty, and *w* is a weight controlling the tradeoff between the cross-entropy loss and adversarial loss. The objective is to find a generator and critic that minimize *L^G^* and *L^C^*, respectively.

### 3.5. Post-Processing

Finally, the 2D MSP images are obtained by post-processing. The post-processing inputs are the 3D mask and original input image. Let *M* be a transformation that represents the correlation of each pixel between *I* and *E*. The illustration is shown in Figure 7. The transformation M is decomposed into two terms as *M* = *TR*, where *R* is a rotation matrix and *T* is a translation matrix. With *R* and *T*, the final 2D MSP images can be obtained.

## 4. Experiments

All of the experimental images were manually labeled by experts through the following steps. The center of the fetal head close to a dark region called the diencephalon was firstly determined and named as the seed point. As shown in Figure 8, the seed point became the origin, and a sagittal plane through the seed point was rotated by *θ_axi_* about the *x*-axis based on the anatomical features on axial planes. Afterward, the plane was rotated by *θ_cor_* about the *y*-axis, corresponding to the coronal view. The rotated plane was the MSP of the fetus and was regarded as the ground truth.

We collected 394 cases of volume data and constructed a database of 3D fetal US images. It is worth mentioning that an improper fetus pose may cause the position of the nuchal translucency to be incorrect, leading to the identification of a defective MSP that is unsuitable for assessing the growth parameter of the nuchal translucency thickness. We utilized oblique angles *θ_axi_* and *θ_cor_* of ±30° as a baseline to determine whether to keep the image. After deleting the cases with poor image quality, tight fetal attachment to the endometrium, and incomplete fetal development, 218 cases of volume data remained for the experiments.

Since the heads of fetuses from two volume data have opposite directions (left and right sides), an alignment step was applied by horizontally flipping the volumes with heads on the right side to the left side. To standardize the dimensions of the training and testing data before feeding them into the model, cubes around the heads of the fetuses were cropped out, which are the most important regions in MSP determination. According to the given seed points coordinates (*x*, *y*, *z*), the cubes were extracted in the range of (*x* ± 40, *y* ± 40, *z* ± 40), resulting in dimensions of 80 × 80 × 80.

For seed point detection, the Adam optimizer was utilized to update the segmentation networks, with a training batch size of 10. The loss function for the segmentation networks was binary cross-entropy. The object detection networks were trained using SGD with 5 × 10^−4^ weight decay and 0.9 momentum, with a training batch size of five. The loss functions for the object detection networks were the cross-entropy and Huber loss. For the proposed GAN, the Adam optimizer was utilized to update the generator and critic, where the batch size was 8, learning rate was 0.0001, *β*_1_ = 0.9, *β*_2_ = 0.999, and $\epsilon =1\;\times 10^{-8}$. Following [19], λ was set to 10. We assigned the weight w as 0.8. The number of total trainable parameters was 4,059,513. The critic and generator were optimized alternately. The proposed framework was developed in Python on an Intel i7 CPU (3.2 GHz, 6 cores) and trained on a single NVIDIA 1080Ti GPU with Tensorflow library from scratch.

We collected 394 cases of volume data and constructed a database of 3D fetal US images. After deleting the cases with poor image quality, tight fetal attachment to the endometrium, and incomplete fetal development, 218 cases of volume data remained for the experiments. Five-fold cross validation was performed on these 218 cases of volume data, and 80% of the data (174 cases) were randomly selected for training and the remaining 20% (28 cases) were used for testing. In the testing phase, only the generator was used to predict a 3D binary mask. In post-processing, the 3D binary mask was processed with the original input image and the final MSP image was obtained.

In total, four metrics were used to evaluate the performance of the proposed network. Given two planes, the manually extracted result ***E***_1_: *a*_1_*x* + *b*_1_*y* + *c*_1_*z* + *d*_1_ = 0 and the predicted result ***E***_2_: *a*_2_*x* + *b*_2_*y* + *c*_2_*z* + *d*_2_ = 0, the first metric is the included angle *θ* between (*a*_1_, *b*_1_, *c*_1_, *d*_1_) and (*a*_2_, *b*_2_, *c*_2_, *d*_2_) (Figure 9a), given by Equation (1):(3)$$\theta =\arccos\left(\frac{\left(a_{1},b_{1},c_{1},d_{1}\right)\cdot \left(a_{2},b_{2},c_{2},d_{2}\right)}{\left\|(a_{1},b_{1},c_{1},d_{1})\right\|\left\|(a_{2},b_{2},c_{2},d_{2})\right\|}\right).$$

The second metric is the Euclidean distance *d* between (*a*_1_, *b*_1_, *c*_1_, *d*_1_) and (*a*_2_, *b*_2_, *c*_2_, *d*_2_), given by Equation (2):(4)$$d=\sqrt{{\left(a_{1}-a_{2}\right)}^{2}+{\left(b_{1}-b_{2}\right)}^{2}+{\left(c_{1}-c_{2}\right)}^{2}+{\left(d_{1}-d_{2}\right)}^{2}}$$

If the two planes coincide with each other, the included angle and Euclidean distance are zero, that is to say, the smaller *θ* and *d*, the better the plane prediction.

For visual comparison, the differences in the yaw and roll angles between the automatically detected and manually extracted MSP were calculated (Figure 9b). The yaw angle *θ_y_* and roll angle *θ_r_* are respectively defined as
(5)$$\theta_{y}=\arctan\frac{b}{a}$$
(6)$$\theta_{r}=\arctan\frac{-c}{\sqrt{a^{2}+b^{2}}},$$
where the equation of the plane is *ax* + *by* + *cz* + *d* = 0

The study was designed with the objective of estimating the variance in automatic and semi-automatic detection. In MSP detection, the mean and variance can be calculated. The mean, standard deviation (SD), and 95% confidence interval of the difference between the automatic and semi-automatic detection results were obtained. Moreover, the association between automatic and semi-automatic detection was assessed by performing a paired sample t-test, wherein *p*
*<* 0.05 was considered statistically significant. We compared the four metrics in five-fold validation by analysis of variance. The statistical analysis was conducted using the Statistical Package for the Social Sciences (SPSS 17.0 for Windows, SPSS Inc., Chicago, IL, USA). Bland–Altman plots were used to assess the bias of the automatic and semi-automatic detection methods.

## 5. Results

The semi-automatic system involved manual determination of the seed points followed by utilizing the GAN-based method to obtain the fetal MSP. The results of the automatic method were obtained by employing the full deep learning method (Figure 10). The execution time of the semi-automatic system was 5 s, while the inference time of the automatic system was about 2.4 s, i.e., up to two times faster than the semi-automatic approach.

The automatic and semi-automatic MSP detection results obtained using the proposed system was compared with the results of manual selection by an expert. The four metrics exhibited no significant differences in five-fold cross-validation. In the automatic system results, 98.6% (*n* = 215) had Euclidean distances less than 0.05, and 89.4% (*n* = 195) of the cases had included angles smaller than 1.0°. The automatic system produced an average included angle of 0.5344° and an average Euclidean distance of 0.0094. The average yaw and roll angles were 0.9253° and 0.1044°, respectively. Most of the cases had small roll and yaw angles simultaneously, meaning that in these cases, the resulting plane could be treated as a sagittal plane (Figure 11). The results reveal that the proposed deep learning method yields conclusions very closed to those obtained by experts.

Table 1 also shows no significant differences between the automatic and semi-automatic MSP detection methods. The high correlation coefficients between the automatic/semi-automatic and manual measurements of the differences in the Euclidean distance, included angle, yaw angle, and roll angle were noted, confirming that the automatic method achieved results consistent with those obtained using the semi-automatic method. Thus, the automatic method can achieve measurement results consistent with those of the semi-automatic method. The differences between the automatic and semi-automatic methods were examined using Bland–Altman plots (Figure 12), and the results of the proposed automatic method agreed well with those of the semi-automatic method.

## 6. Discussion

In the first trimester, the MSP has proven to be useful for assessing fetal development and congenital fetal anomalies [1]. The optimal plane acquired in prenatal US is important for obtaining valid, precise, and reproducible measurements [8,22]. Expert training is required to achieve high quality examination. Therefore, learning-based methods, such as convolutional neural networks, have been utilized in the second trimester of pregnancy [23,24,25]. In the present study, we developed an accurate automatic system using deep learning to help resolve the problems encountered in conventional manual, two-dimensional (2D) methods [15]. We proposed not only a GAN-based method of fetal MSP detection from 3D US images, but also a deep learning method to obtain an exact seed point. To the best of our knowledge, the proposed system using an automatic GAN-based approach for fetal MSP detection is the first to be introduced.

Although some semi-automatic and automatic systems involving 2D US have been developed for first trimester fetal evaluation [16,17,22,26], we presented a novel automatic MSP detection system with excellent accuracy. Our automatic MSP detection system is the most precise system thus far for fetal MSP evaluation in the first trimester of pregnancy.

The results presented in this report validate the automatic fetal MSP detection approach using 3D US and provide evidence of its potential clinical applicability. The fetal structures, such as the nuchal translucency, nose tip, and translucent diencephalon, could be measured in the proposed system based on the exactly detected MSP. Moreover, the experimental results obtained using the proposed method and the corresponding evaluations demonstrate its consistency with manual measurements and potential for routine clinical usage. We believe that the overall trade-off between time and accuracy is acceptable.

The proposed automatic method of fetal MSP detection from 3D US images based on a GAN treats MSP detection as a filtration problem, where the neural network is used as a filter to generate 3D masks that contain the information about the plane position. Moreover, the proposed deep learning method enables the exact initial seed point to be found, serving as a reference for the subsequent filtration. By using the transformation of the initial and estimated planes, the post-processing provides the final MSP. The experimental results of five-fold cross validation reveal that the proposed system can deal with the MSP detection problem and achieves good performance.

The advantage of the proposed system is that full deep learning using the GAN can be performed without any user interaction in a short time. The average time for manual evaluation depends on the clinical condition of the fetus and the experience of the clinician. It usually takes a few minutes. The average execution time is 2.4 s per image, while manual measurement is time consuming due to the aforementioned difficulties of US examination. The proposed approach is also up to two times faster than the semi-automatic method.

A limitation of the system is that when the fetus moves or has other soft tissues adhered to it during 3D US acquisition; the image analysis becomes complicated, making it difficult to retrieve a complete set of measurements. Furthermore, the image retrieved from smaller fetus could escalate the error of image processing. Moreover, poor US image quality caused by speckle noise, fuzzy boundaries, and weak edges increases the difficulty of the deep learning progress.

Establishing the fetal MSP accurately using our automatic system will enable the difficulties in implementing important markers during the first trimester to be overcome. We believe that automatic detection of the fetal MSP is clinically useful and that our proposed system may be usefully applied to other clinical fields in the future.

## 7. Conclusions

This approach not only preserves the 2D and 3D geometry simultaneously, but also seeks the answer directly rather than requiring a complicated transformation procedure. To the best of our knowledge, no automatic GAN-based fetal MSP detection method has been introduced previously. Moreover, the execution time for one case using the proposed method is considerably improved compared to those obtained in previous works, increasing the efficiency and reducing the intra- and inter-observer variability. The automatic system can successfully detect fetal MSPs in 3D US images, which can reduce the assessment time, increase the accuracy, and enhance professional training. This method could also solve clinical dilemmas by shortening training time and improving training quality.

## Figures and Tables

**Figure 1 diagnostics-11-00021-f001:**
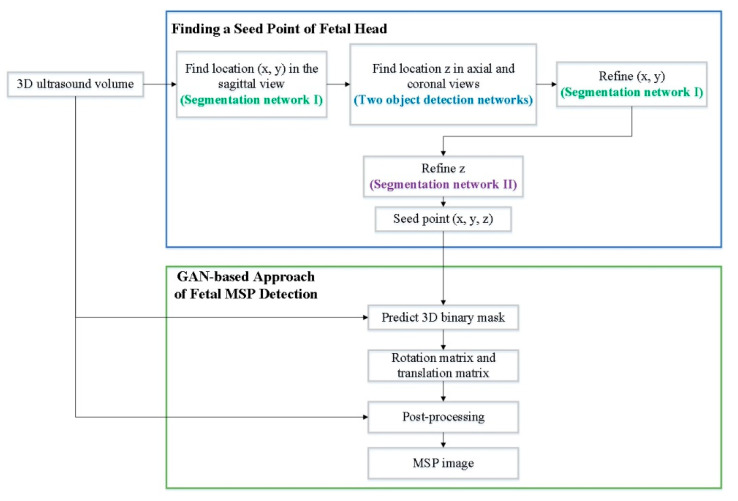
Flow diagram of automatic middle sagittal plane (MSP) detection using generative adversarial networks (GANs).

**Figure 2 diagnostics-11-00021-f002:**
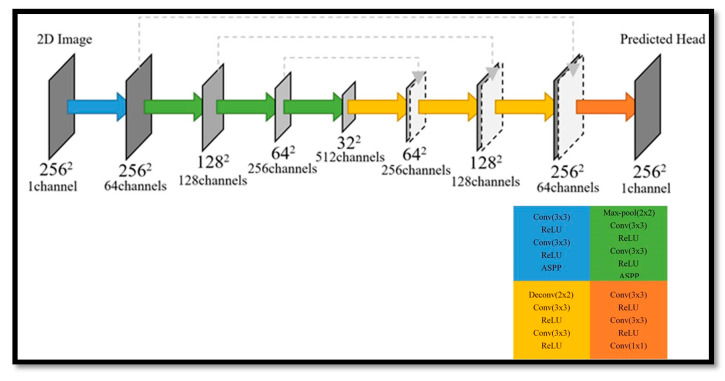
Illustration of Unet + ASPP architecture.

**Figure 3 diagnostics-11-00021-f003:**
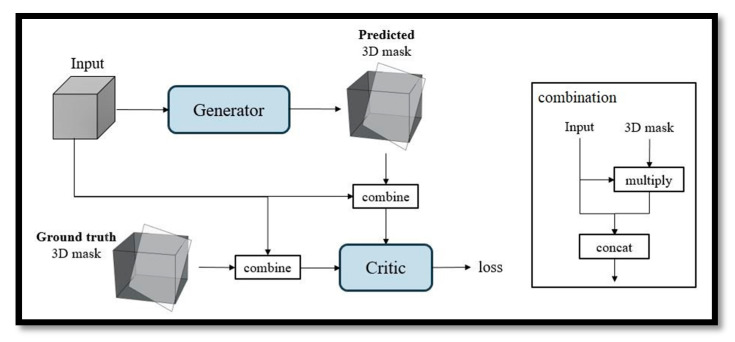
Training phase.

**Figure 4 diagnostics-11-00021-f004:**
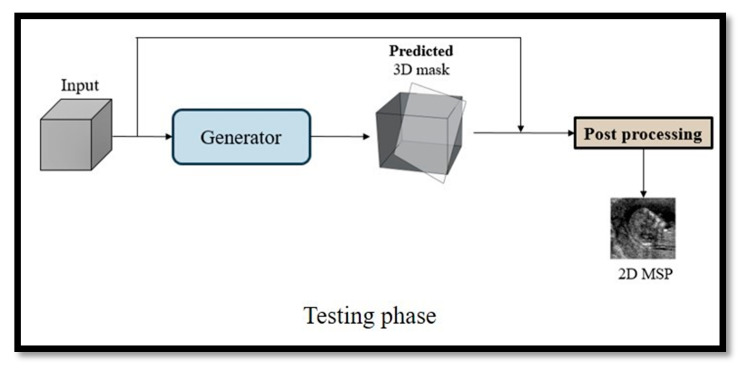
Testing phase.

**Figure 5 diagnostics-11-00021-f005:**
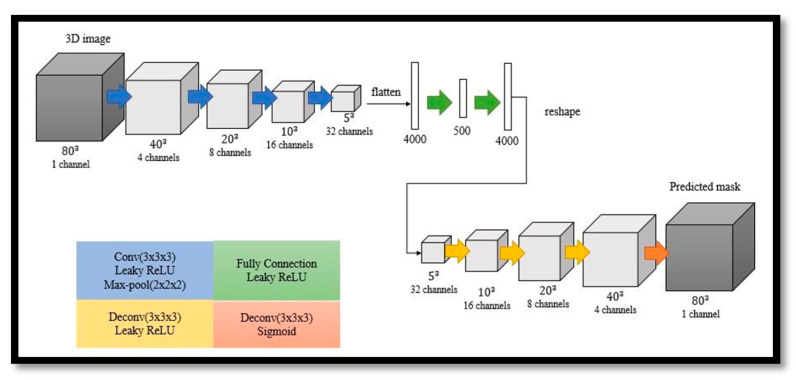
Network architecture of generator.

**Figure 6 diagnostics-11-00021-f006:**
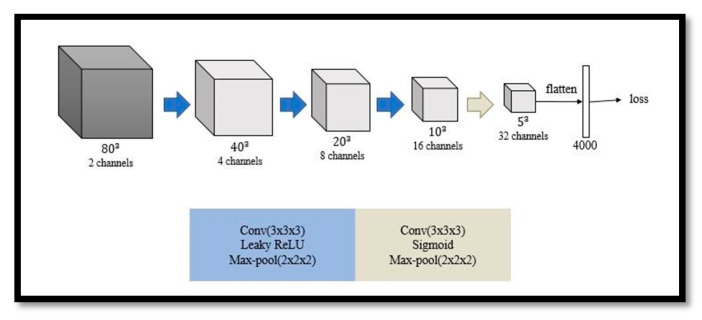
Network architecture of critic where Leaky ReLU is a modified ReLU function to allow positive values and small negative values.

**Figure 7 diagnostics-11-00021-f007:**
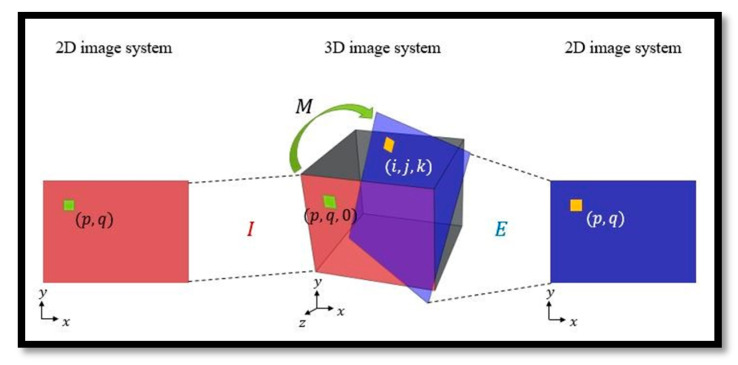
Illustration of the transformation between two planes where *M* is a transformation and *I* is the initial sagittal plane. Each pixel (*p*, *q*) of *I*, i.e., a voxel (*p*, *q*, 0) in the 3D image space, is transformed to (*i*, *j*, *k*) on *E* through *M* whose intensity value is mapped onto the corresponding coordinate (*p*, *q*) of *E*.

**Figure 8 diagnostics-11-00021-f008:**
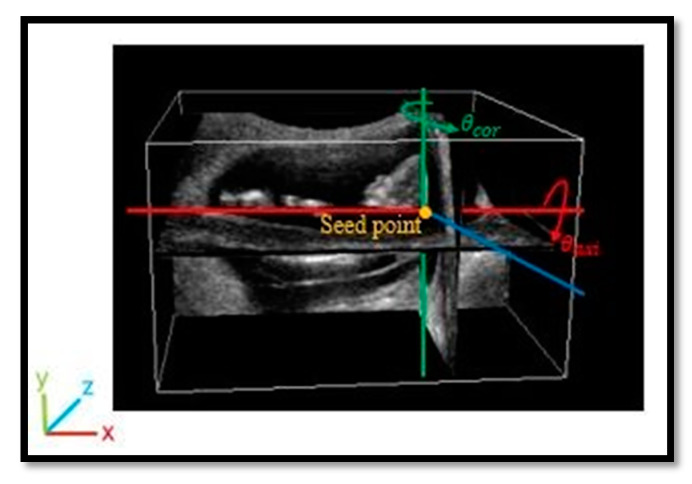
Illustration of labelling.

**Figure 9 diagnostics-11-00021-f009:**
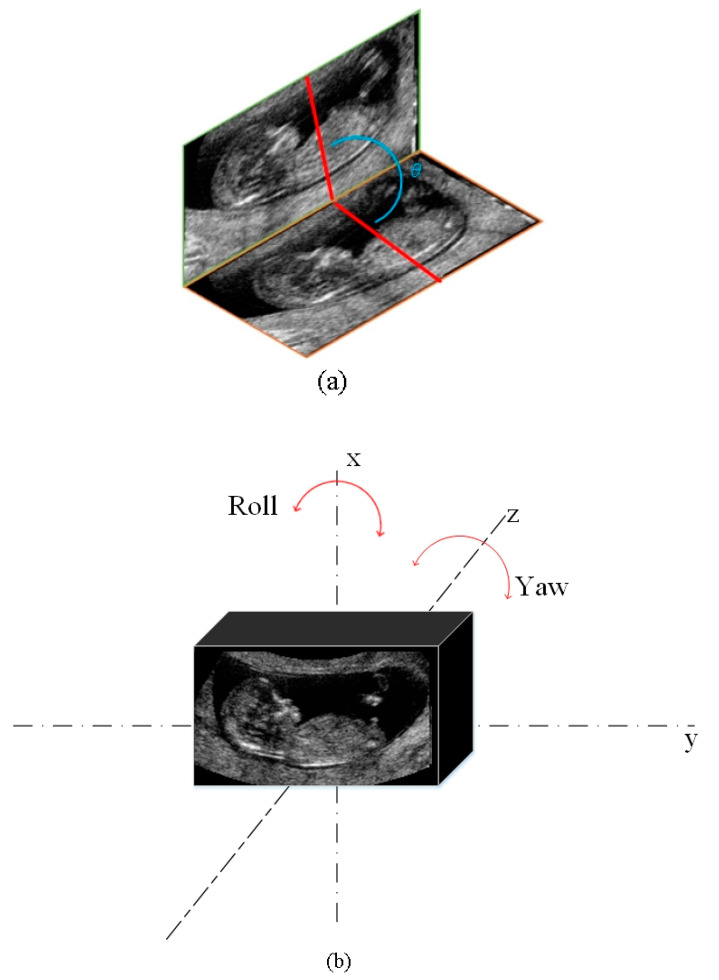
Metrics to evaluate the performance of the proposed network. (**a**) The included angle θ is the angle between the automatically detected and manually extracted MSPs. (**b**) The roll and yaw angles indicate the rotation of the detected MSP with respect to the x- and z-axes, respectively.

**Figure 10 diagnostics-11-00021-f010:**
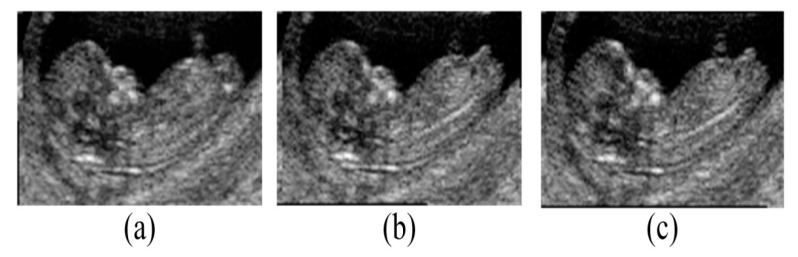
Midsagittal plane extraction using the (**a**) manual, (**b**) semi-automatic, and (**c**) automatic methods.

**Figure 11 diagnostics-11-00021-f011:**
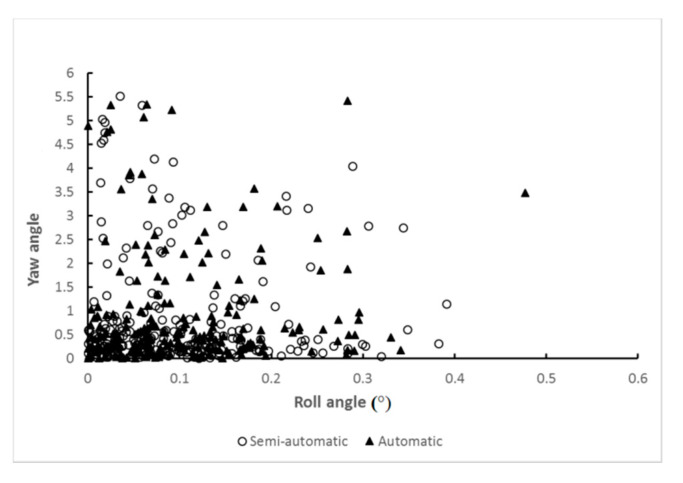
Consistency of the yaw and roll angle results. Each point represents a case, where the x- and y-axes are the roll and yaw angles, respectively (circles: semi-automatic; triangles: automatic).

**Figure 12 diagnostics-11-00021-f012:**
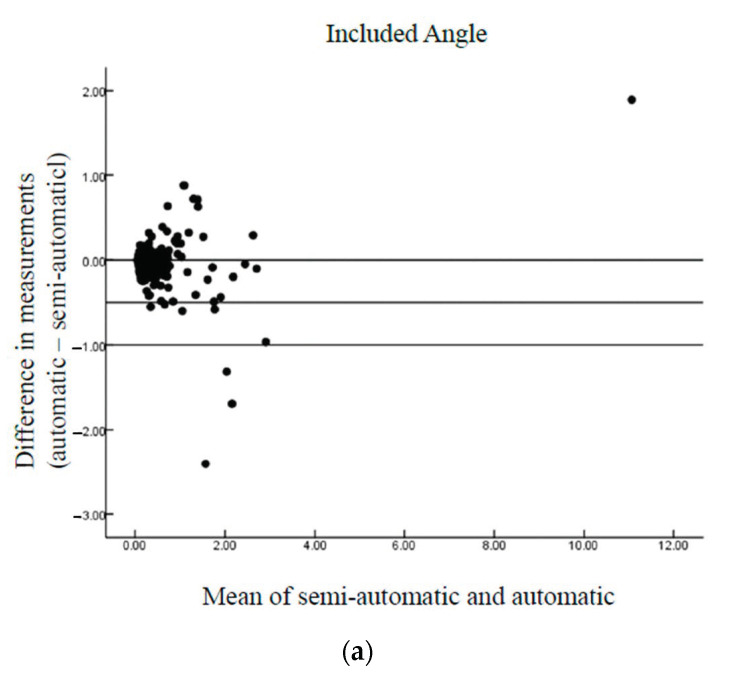
Bland–Altman plots of the differences between automatic/semi-automatic and manual methods of fetal MSP detection: (**a**) included angle, (**b**) Euclidean distance, (**c**) yaw angle, and (**d**) roll angle. The lines indicate the mean bias and 95% limits of agreement.

**Table 1 diagnostics-11-00021-t001:** Comparison of the automatic/semi-automatic and manual fetal MSP detection methods.

Voxel						
95% CI of Difference						
Type	Mean	SD	Lower	Upper	*r*	*P*
Angle						
Semi-automatic	0.4951	0.9278	−0.0833	0.0048	0.9346	0.648
Automatic	0.5344	0.8671				
Euc-distance						
Semi-automatic	0.0087	0.0166	−0.0015	0.0001	0.9368	0.6588
Automatic	0.0094	0.0154				
Yaw						
Semi-automatic	0.9057	1.2072	−0.1355	0.0963	0.7394	0.8651
Automatic	0.9253	1.1972				
Roll						
Semi-automatic	0.1004	0.0829	−0.0123	0.0043	0.7142	0.6156
Automatic	0.1044	0.0817				

## Data Availability

The data presented in this study are available on request from the corresponding author. The data are not publicly available due to privacy.

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
