# Peer review of "Automatic Fetal Middle Sagittal Plane Detection in Ultrasound Using Generative Adversarial Network"

_diagnostics, 2020, doi:10.3390/diagnostics11010021_

Round 1
Reviewer 1 Report
- Why only 3DUS was considered for the screening? It is known that, even though 3DUS image quality has improved over the last years, 2DUS still provides, in general, better image quality overall, remaining the preferred image acquisition method. The authors should better defend the use of 3DUS imaging over 2D.
- It would be interesting to see how these automatic plane detection approaches work with abnormal cases. Since only cases without any congenital anomalies were recruited, the authors would need to comment on this.
- “The segmentation network firstly finds the seed point int he sagittal view […] two object detection networks are used to identify location z in the axial and coronal views” —> These are fetal US images, where one of the big challenges is the random position-location of the fetus. Therefore, it is very hard to identify the actual sagittal, coronal, and/or axial planes. How is the volume oriented?
- Line 96 —> Typically, in the context of DL terminology, it is called “discriminator” not “critic”. Please, replace it in the entire manuscript.
- Lines 115 - 118: Actually, classifying all possible planes is not that hard, computationally speaking. The true challenge is the orientation of the fetus-volume, so these planes can be extracted.
- Line 119: “we employed a deep learning method to find a seed point of the fetal head and a neural network to generate a 3D binary mask” —> The first DL model is actually a neural network. Please, use the terms properly.
- Line 127: “The two object detection networks are deeply supervised object detectors” —> I’m not sure what this sentence means. Please, clarify.
- Please, provide references for the UNet networks. Also, please, define what ASPP means.
- It is still unclear to me how the first network works. Perhaps, a more illustrative figure would be of help.
- There are some parts of the manuscript that repeat the same text. For instance, when describing how the GAN network works at the begining of section 1.2.
Line 214 —> the total number of volumes and the remaining ones was just explained in the previous paragraph.
- Please, provide the details of the human tasks: how many clinicians, were these the same that defined the ground-truth, what is their speciality, and degree of expertise, etc.
- Typically, one of the biggest challenges in fetal imaging is the orientation of the fetus body, which makes any automation process very complicated. Was any acquisition protocol defined when acquiring the 3DUS volumes? How much variety is the proposed framework able to deal with? Does it work with any random 3DUS volume, as long as the fetus is inside the acquired volume?
Please, clarify.
- Line 359: Do the authors have an estimation of the average time for manual acquisition-evaluation?
Author Response
Reviewer 1
- Why only 3DUS was considered for the screening? It is known that, even though 3DUS image quality has improved over the last years, 2DUS still provides, in general, better image quality overall, remaining the preferred image acquisition method. The authors should better defend the use of 3DUS imaging over 2D.
[Answer] Thanks for your question. 2D US does provide many useful images, but the ability to achieve a reliable 2D US image is dependent on appropriate training and adherence to a standard technique to achieve uniformity of results among different operators. Although 3D US image provides lots of useful information, it still could not replace the 2D US. However, in the automatic algorithm, 3D US can provide complete image volume data for complete analysis.
- It would be interesting to see how these automatic plane detection approaches work with abnormal cases. Since only cases without any congenital anomalies were recruited, the authors would need to comment on this.
[Answer] Thanks for your comments. In the future, we hope this method could apply in clinical practice and the detection of the abnormal fetus is necessary. However, in this stage of system development, we only recruited normal cases to maintain system stability. We will use this system to detect abnormal cases and evaluate the efficacy.
- “The segmentation network firstly finds the seed point int he sagittal view […] two object detection networks are used to identify location z in the axial and coronal views” —> These are fetal US images, where one of the big challenges is the random position-location of the fetus. Therefore, it is very hard to identify the actual sagittal, coronal, and/or axial planes. How is the volume oriented?
[Answer] Thanks for your question. As shown in below Figure, the heads of fetuses from two volume data have opposite directions, left side and right side. To regularize the direction of fetuses, an alignment step is applied by horizontally flipping the volumes with heads on the right side to the left side.
(a) The sagittal plane (1) |
(b) The sagittal plane (2) |
(c) The axial plane (1) |
(d) The axial plane (2) |
(e) The coronal plane (1) |
(f) The coronal plane (2) |
- Line 96 —> Typically, in the context of DL terminology, it is called “discriminator” not “critic”. Please, replace it in the entire manuscript.
[Answer] Thank you for pointing it. In the work of WGAN, Arjovsky et al. [1] rename the discriminator to critic for emphasizing its property. In this paper, we also call discriminator as critic. We have modified in Line 103-105.
- Lines 115 - 118: Actually, classifying all possible planes is not that hard, computationally speaking. The true challenge is the orientation of the fetus-volume, so these planes can be extracted.
[Answer]
The orientation of the fetus-volume is aligned by preprocessing step in a manual operation.
- Line 119: “we employed a deep learning method to find a seed point of the fetal head and a neural network to generate a 3D binary mask” —> The first DL model is actually a neural network. Please, use the terms properly.
[Answer] Thank you for the suggestion. The sentence is modified according to your suggestion.
- Line 127: “The two object detection networks are deeply supervised object detectors” —> I’m not sure what this sentence means. Please, clarify.
[Answer] The two object detection networks are deep learning networks that are used to modify the predicted seed point. The sentence has modified in the revised manuscript.
- Please, provide references for the UNet networks. Also, please, define what ASPP means.
[Answer]
The references have been added in the revised manuscript and the definition of ASPP has also added in the revised manuscript. We have modified the manuscript in Line 134-141.
- It is still unclear to me how the first network works. Perhaps, a more illustrative figure would be of help.
[Answer]
We have modified Figure 1 to illustrate that how the first network works.
- There are some parts of the manuscript that repeat the same text. For instance, when describing how the GAN network works at the begining of section 1.2.
Line 214 —> the total number of volumes and the remaining ones was just explained in the previous paragraph.
[Answer] Thanks for your comments. We have improved our manuscript to delete the repeat text.
- Please, provide the details of the human tasks: how many clinicians, were these the same that defined the ground-truth, what is their speciality, and degree of expertise, etc.
[Answer] Thanks for your suggestion. We have provided the details of the human tasks in line 85 as following: The fetal MSP was detected automatically using the software and manually by the obstetrics doctor with 20 years of experience (one of the authors, P.Y. Tsai MD.).
- Typically, one of the biggest challenges in fetal imaging is the orientation of the fetus body, which makes any automation process very complicated. Was any acquisition protocol defined when acquiring the 3DUS volumes? How much variety is the proposed framework able to deal with? Does it work with any random 3DUS volume, as long as the fetus is inside the acquired volume? Please, clarify.
[Answer] Thanks for your question. The image volumes were acquired by the appropriate training of sonographers and adherence to a standard technique in accordance with the guidelines that were established by The Fetal Medicine Foundation (FMF). The guidelines of 3D US include the whole fetus (fetal crown-rump length should be obtained), the fetus is in the neutral position, and the amnion is seen separately from the nuchal membrane. And we also have modified the manuscript in Line 77-82.
- Line 359: Do the authors have an estimation of the average time for manual acquisition-evaluation?
[Answer] Thanks for your question. The average time for manual evaluation depends on the clinical condition of the fetus and the experience of the clinician. It usually takes a few minutes. The average execution time of this system is 2.4 seconds per image, while manual measurement is time consuming due to the aforementioned difficulties of US examination. We also modified our manuscript in Line 361-363.
Reference
- Arjovsky, M.; Chintala, S.; Bottou, L. Wasserstein gan. arXiv preprint arXiv:1701.07875 2017.

Reviewer 2 Report
Interesting work with clinical relevance.
The authors should explain more clearly that the mid sagittal plane is the basis for the nuchal translucency exam that provides a risk assessment for chromosomal aberrations and is performed between 11-14 weeks of gestation.
Were measurements of nuchal translucency performed ? Were these measurements compared to manual measurements?
I agree with the limitations the authors have mentioned. In addition, a major factor in 3D imaging especially when performed trans abdominally is maternal habitus, prior cesarean section etc. If you have data regarding the performance of your method in regard for example to maternal BMI, this is a clinically relevant data.
Author Response
Interesting work with clinical relevance.
The authors should explain more clearly that the mid sagittal plane is the basis for the nuchal translucency exam that provides a risk assessment for chromosomal aberrations and is performed between 11-14 weeks of gestation.
[Answer] Thanks for the comment. We modified the introduction in Line 51-52 and explain more clearly that the mid sagittal plane is the basis for the nuchal translucency exam that provides a risk assessment between 11-14 weeks of gestation.
Were measurements of nuchal translucency performed? Were these measurements compared to manual measurements?
[Answer] Thanks for your question. We plan to develop the other automatic method to measure the nuchal translucency in the future and also compared to the manual method.
I agree with the limitations the authors have mentioned. In addition, a major factor in 3D imaging especially when performed trans abdominally is maternal habitus, prior cesarean section etc. If you have data regarding the performance of your method in regard for example to maternal BMI, this is a clinically relevant data.
[Answer] Thanks for your comments. Indeed, maternal factors affect the image quality of trans-abdominal ultrasound. However, we didn’t analyze the relationship between clinical data and image limitation. In this study, we focused on the image analysis and we’ll emphasize this intelligent suggestion in the future work.